# Clinoptilolite—A Sustainable Material for the Removal of Bisphenol A from Water

Alina Marilena Dura [1], Daniela Simina Stefan [1,*], Florentina Laura Chiriac [2], Roxana Trusca [3], Adrian Ionut Nicoara [1] and Mircea Stefan [4,*]

1 Faculty of Chemical Engineering and Biotechnologies, National University of Science and Technology POLITEHNICA Bucharest, 1-7 Polizu Street, 011061 Bucharest, Romania; pahontu_alina@yahoo.com (A.M.D.); adrian.nicoara@upb.ro (A.I.N.)

2 National Institute for Research and Development for Industrial Ecology—INCD ECOIND, 57-73 Drumul Podu Dambovitei Street, District 6, 060652 Bucharest, Romania; laura.badea88@yahoo.com

3 Faculty of Engineering in Foreign Languages (FILS), National University of Science and Technology POLITEHNICA Bucharest, 313 Splaiul Independenței Street, 011061 Bucharest, Romania; truscaroxana@yahoo.com

4 Pharmacy Faculty, University Titu Maiorescu, No. 22 Dâmbovnicului Street, District 4, 040441 Bucharest, Romania

* Correspondence: daniela.stefan@upb.ro (D.S.S.); stefan.mircea@incas.ro (M.S.)

**Abstract:** Bisphenol A is a remarkable chemical compound as it has many applications, mainly in the plastics industry, but it also has toxic effects on the environment and human health. This article presents a comparative study regarding the adsorption of BPA on Active carbon and zeolitic tuff, ZTC. In this paper, the characterization of the zeolitic tuff, adsorbent, was carried out from an elemental and mineralogical point of view, and it noted the pore size and elemental distribution, using SEM, EDAX, and XRD analysis. The pore size varies from 30 nm to 10 μm, the atomic ratio is $Si/Al \geq 4$, and 80% of the mineralogical composition represents Ca Clinoptilolite zeolites and Ca Clinoptilolite zeolites $((Na_{1.32}K_{1.28}Ca_{1.72}Mg_{0.52})$ $(Al_{6.77}Si_{29.23}O_{72})(H_2O)_{26.84})$. Moreover, a comparative study of the adsorption capacity of bisphenol A, using synthetic solutions on an activated carbon type—Norit GAC 830 W, GAC—as well as on Clinoptilolite-type zeolitic tuff—ZTC, was carried out. The experiments were carried out at a temperature of 20 °C, a pH of 4.11, 6.98, and 8.12, and the ionic strength was assured using 0.01 M and 0.1 M of KCl. The adsorption capacities of GAC and ZTC were 115 mg/g and 50 mg/g, respectively, at an 8.12 pH, and an ionic strength of 0 M. The Langmuir mathematical model best describes the adsorption equilibrium of BPA. The maximum adsorption capacity for both adsorbents increased with an increasing pH, and it decreased with increasing ionic strength.

**Keywords:** bisphenol A; adsorption; activated carbon; zeolite clinoptilolite; sustainable materials; water treatment

## 1. Introduction

Bisphenol A, BPA, (4,4-isopropylidenediphenol), is an anthropogenic compound that is moderately soluble in water (120–300 mg/L) at room temperature, and highly soluble in alkaline solutions, such as ethanol and acetone [1,2].

Over 3.8 million tones of BPA is produced each year, and it is mainly used as a monomer to produce polycarbonate, a precursor of epoxy resins [3], and vinyl ester resins [4]. Plastic packaging that comes into contact with water and food is a source of BPA in household wastewater and natural water sources [5]. BPA, a persistent organic pollutant present in various types of water, is confirmed as an endocrine disruptor, therefore, it is essential that it is effectively removed from the environment, to protect both the natural environment and human health [6,7]. Following the evaluation of the BPA

pollution level, the European Union decided to introduce the Drinking Water Directive in 2020 (2020/2184), which aims to protect the quality of drinking water. This directive introduced the obligation to monitor and reduce the content of BPA in drinking water to a level lower than 2.5 micrograms/L.

In this context, technological processes must be identified to include the use of materials or methods aimed at removing BPA to the level required by the legislation in force.

Researchers have studied and used different methods/techniques to remove BPA contained in wastewater. They found that treating BPA is difficult with conventional wastewater treatment methods because of its structure, which allows molecules to escape the primary and secondary treatment facilities. Thus, there is a need to use advanced BPA removal techniques, which include physicochemical or enzymatic methods [8,9], advanced oxidation [10,11], photocatalysis [12,13], ultrasonic degradation [14,15], and photodegradation [16].

To evaluate the efficiency of BPA degradation, rapid analysis methods for BPA monitoring are essential. The techniques that allow for the monitoring of the BPA removal time during different water treatment processes are as follows: HPLC (high performance liquid chromatography), LC/MS (liquid chromatography/ass spectrometry), and capillary electrophoresis [17].

This article presents the latest investigations that are aimed at the removal and degradation of BPA via adsorption. Depending on the interaction of the adsorbate with the adsorbent surface, adsorption can be physical, as in the case of the formation of weak van der Walls bonds between the adsorbate and the adsorbent surface, or chemical, which are either strong ionic or covalent bonds [18].

Adsorption is an effective technique for removing BPA from effluents because it is low cost, environmentally friendly, it uses a wide range of adsorbents with high reusability, and it is an easy operation [19,20]. The adsorbate is retained on the adsorbent, which is the material that performs the adsorption (solid/liquid). The use of adsorbents has the advantage of retaining substances in low concentrations, and they can select certain substances. Adsorbents can be inorganic or organic materials, and they are porous substances with high specific surface areas. Inorganic materials such as clay minerals, zeolites, and nanomaterials were used for BPA adsorption, and activated carbon, graphene, polymers, agricultural waste were used as organic materials [19].

Porous carbonaceous materials (PCMs) have a large number of interconnected pores throughout the matrix, and their properties, such as chemical stability, large surface area, easy processability, and hierarchical porosity, make them interesting research subjects. Porous carbonaceous materials have different forms, such as soft and hard mesoporous carbons, porous nanocarbons, activated carbon, and heteroatom-doped mesoporous carbons [21]. Adsorption on activated carbon has been investigated for pollutant removal, due to its high performance and low cost.

Activated carbon produced from shrimp shells, generated by seafood industries, has a large specific surface area and abundant active sites; the production process is also low cost. The adsorption capacity of activated carbon is affected by the preparation conditions, as follows: heating rate and time and gas flow. However, various agents can be used to improve the structure and increase the specific surface area during the preparation of this adsorbent. $CO_2$ and $NaHCO_3$ are activated carbon activators, which favor physical activation by removing the blockage of the activated carbon surface, and increasing the adsorption capacity, respectively. This occurs by increasing the porosity, specific surface area, and functional groups on the carbon surface. Magnetically activated carbon is created by doping magnetic elements into activated carbon during pyrolysis, which can be easily separated using a magnetic field. The researchers determined a maximum adsorption performance (98.01%) of BPA when used on SS@C.AC-M, obtained at pH 2.0, at an initial BPA concentration of 25 mg $L^{-1}$ [22].

The use of pretreated activated carbon fibers for BPA adsorption resulted in the 98–99.9% removal of BPA in the aqueous phase, under the optimal conditions of pH 7,

at 15 °C, for 2 min. The process follows the second-order kinetic model and Langmuir adsorption isotherms, as determined by the researchers [23]. In the presence of NaCl, CaCl$_2$, and MgCl$_2$ ionic salts, ACF exhibits a maximum BPA adsorption of 100 mg/L for MgCl$_2$, and 10 mg/L for NaCl and CaCl$_2$.

Zeolites exhibit excellent thermal stability, a good generation performance, incombustibility, and a high adsorption capacity. Hydrophobicity is required for zeolites to remove hydrophobic pollutants. Researchers have developed methods to transform the hydrophilicity of zeolites into hydrophobicity by patching a hydrophobic organic polymer layer onto zeolites, followed by ion exchange, the anchoring of organic groups and inorganic functional groups, acid leaching, and finally, calcination [24]. In recent years, a wide variety of surfactant-modified β-cyclodextrin and Cu/Fe bimetallic zeolites, with improved adsorption capacities and cost-effectiveness, have been developed.

The efficient removal of BPA from aqueous solutions was investigated by researchers using a zeolite imidazole framework, which demonstrated a good adsorption capacity of BPA molecules. Highly porous ZIF-8 was used in the experiments that included the following factors: BPA concentrations, pH, doses of ZIF-8, and contact time. A combination of these factors led to the best performance, with a BPA removal efficiency of 99.93% [25].

Surfactant-modified zeolites show high BPA adsorption. BPA was adsorbed on natural zeolite modified with the cationic surfactant didodecyldimethylammonium bromide, with and without pretreatment of the zeolite with NaCl and HCl. BPA molecules can be adsorbed using the modified zeolite via interactions with the zeolite surface. This occurs when bonds are made between the metal atoms in the zeolite, the surfactant, and the oxygen atom in the OH group of BPA. The adsorbate–adsorbent interaction mechanism may include electrostatic attraction, hydrophobic interactions with the surfactant, and chemisorption [26].

NaX synthetic zeolites modified with β-cyclodextrin can improve BPA adsorption. The adsorption kinetics of BPA followed the pseudo-second order model, indicating electron exchange. BPA adsorption fit to the Langmuir isotherm, monolayer adsorption, and hydrogen bonding to form host–guest complexes. The maximum adsorption capacity for BPA was 32.7 mg/g, indicating that NaX–CD can be an effective adsorbent, in accordance with Ref. [27]. BPA was effectively removed using T-Hβ (25, 50) zeolites, which showed a high adsorption capacity because their pore size is larger than the size of the BPA molecule. The adsorption process follows the Redlich–Peterson model, and the maximum adsorption capacity had a value of 117.62 mg/g [24].

In this paper, the characterization of the zeolitic tuff-type, Clinoptilolite—, ZTC, from Rupea, Romania, —as an adsorbent, was carried out from an elemental and mineralogical point of view, wherein pore size and element distribution were taken into consideration. A comparative study of the removal efficiency of bisphenol A, using synthetic solutions on an activated carbon type—Norit GAC 830 W, GAC—and on Clinoptilolite-type zeolitic tuff— ZTC, from Rupea, Romania —was also carried out.

The adsorption isotherms were drawn, and the mathematical models that best describe the adsorption process, as well as the mechanisms underlying the adsorption process, were identified.

The influence of Ci, pH, and ionic strength on the adsorption capacity of the adsorbents used was studied.

Considering the harmful effect of BPA on life, it is essential that it is used as little as possible, and that the most efficient, cost-effective, and environmentally friendly techniques are used to remove it, degrade it efficiently and quickly, and without negative side effects.

## 2. Materials and Methods

### 2.1. Chemicals and Equipments

Chemical reagents used were provided by Sigma Aldrich, as follows: bidphenol A solution 99%, methanol, hydrochloric acid, glycol, diacid potassium phosphate and monoacid sodium phosphate, and potassium chloride. The adsorbent materials used

in the study are activated carbon type—Norit GAC 830 W, GAC—and zeolitic tuff type Clinoptilolite—ZTC—produced by the Zeolites Development SRL, Rupea Romania, both of which are approved in accordance with the hygiene requirements of the drinking water standard, EN 12915/2003.

The characteristics of the activated charcoal, Norit GAC 830 W, are shown in Table 1.

**Table 1.** The technical specification (datasheet) of activated carbon type, Norit GAC 830 W, GAC [28].

| No Crt. | Specification | Active Charcoal Norit GAC 830 W |
|---|---|---|
| 1 | Particle size > 2.36 mm | Max 15% in mass unit |
| 2 | Particle size < 0.6 mm | Max 5% in mass unit |
| 3 | Moisture | Max 5% |
| 4. | Iodine number | 957 |
| 5 | Methylene blue adsorption | 20 g/100 g |
| 6 | Ash content | 12% |
| 7 | Total surface area, BET analises | 1100 m$^2$/g |
| 8 | Apparent density | 500 kg/m$^3$ |

To maintain the pH at constant values of 4.11, 6.98, and 8.12, buffer solutions prepared from hydrochloric acid and glycol, and diacid potassium phosphate and monoacid sodium phosphate, were used in well-established ratios, respectively. The ionic strength values were assured using KCl.

The concentrations of bisphenol A, BPA, were determined using a high-performance liquid chromatograph (HPLC) from the company Agilent, Series 1100, Agilent Technologies, Santa Clara, CA, USA and it was equipped with a UV-DAD detector. The investigation of the ZTC was carried out with the help of the scanning electron microscope, Quanta Inspect F50, FEI Company, Eindhoven, Netherlands, which was equipped with a field emission electron gun—FEG (field emission gun)—with a resolution of 1.2 nm, and an energy dispersive X-ray spectrometer (EDS) with a resolution of MnK of 133 eV.

For the investigation of the samples, and for good conduction from an electrical point of view, the samples were metallized for 60 s with gold.

The structural information regarding ZTC was obtained via the X-ray diffraction (XRD) technique, which was carried out in air and at room temperature, with the help of PANalytical Empyrean (Almelo, Netherlands) equipment that comprised a characteristic Cu X-ray tube ($\lambda$ Cu K$\alpha$1 = 1.541874 Å). Two samples (P0 and P1) were scanned in a 2$\theta$ angle range of 10–80°, with a scan increment of 0.02°, and a time of 100 s/step. Phase identification and Rietveld quantitative phase analyses were performed, using X'Pert High Score Plus 3.0 software (PANalytical, Almelo, The Netherlands). The other instruments used are as follows: Jenway 370 pH-meter, Jenway Scientific Instruments, Essex, United Kingdom, analytical balance Precisa type XB 120 A, Precisa company, Dietikon, Switzerland, and analog orbital shaker Velp, VELP Scientifica, Usmate, Italy.

## 2.2. Preparation of Samples

Synthetic solutions of BPA with concentrations between 1–300 mg/L were used. The stock solution was prepared by weighing 0.1 g of solid BPA, then it was dissolved in ethanol. Distilled water was added until the required volume of solution was obtained. The ratio of solid/liquid in the systems was 1 g adsorbent/1 L solution. The experiments were carried out at a temperature of 20 °C, and a pH of 4.11, 6.98, and 8.12. To maintain the pH at a preset value, buffer solutions of hydrochloric acid and glycol were used at a pH of 4.11, and diacid potassium phosphate and monoacid sodium phosphate were used in a well-established ratio, at pH values of 6.98 and 8.11. The ionic strength was assured using KCl, and the final concentrations in the studied sites were 0.01 M and 0.1 M.

The adsorption capacity ($q_e$) of the adsorbents was calculated as follows [18]:

$$q_e(\text{mg/g}) = \frac{C_0 - C_e}{m} \cdot V \tag{1}$$

where $C_0$ and $C_e$ represent the initial and equilibrium concentration of the BPA solution mg/L, $V$ is the volume of the solution, L, and $m$ is the mass of the adsorbent, g.

### 2.3. Adsorption Isotherms

A comparative study concerning experimentally obtained isotherms with Langmuir and Freundlich type isotherms for BPA on GAC and ZTC was conducted.

### 2.3.1. Langmuir Type Isotherm

The experimentally obtained isotherms were compared with theoretical Langmuir type and empirical Freundlich type isotherms [23,29].

The characteristic equation of the Langmuir isotherm is as follows:

$$q_e = \frac{bq_m Ce}{1 + bCe} \tag{2}$$

where $q$ = adsorption capacity at equilibrium, mmolg/g; $q_m$ = maximum adsorption capacity for a certain set of conditions at equilibrium, when the entire monomolecular layer is occupied, mmol/g; $Ce$ = concentration of the solute in the system at equilibrium, mmol/L; and $b$ = constant that depends on the nature of the system, the equilibrium constant, and the adsorption coefficient.

The Langmuir equation can be written in linearized form, as follows:

$$\frac{1}{q_e} = \frac{1}{q_m} + \frac{1}{bq_m Ce} \tag{3}$$

The constant, $b$, and the maximum adsorption capacity, "$q_m$", can be determined from experimental data, if $1/q$ is graphically represented as a function of $1/Ce$.

The graphical representation is a straight line that intersects Oy at the point $(0.1/q_m)$, from which, $q_m$ can be determined. If $q_m$ is known, the "$b$" is determined from the value of the tangent of the angle that the line makes with the Ox axis.

### 2.3.2. Freundlich Type Isotherm

The characteristic equation of the Freundlich isotherm is as follows [23,29]:

$$q_e = KC_e^{1/n} \tag{4}$$

where $n$ and $K$ represent constants that are specific to each system, and which depend on the working temperature. The constants $K$ and $n$ can be determined from experimental data, if represented graphically, as follows:

$$\log q_e = \text{f}(\log C_e) \tag{5}$$

A straight line is obtained when this equation is used. The intersection with Oy is $\log K$, and the tangent of the angle formed with the abscissa is $1/n$. From here, the two constants for $K$ and $n$ are obtained; then, the isotherm is drawn.

The linearization of the equation is:

$$\log q_e = \log K + (1/n) \log C_e \tag{6}$$

### 3. Results

*3.1. ZTC Adsorbant Characterisation*

3.1.1. SEM and EDAX Analysis for ZTC

The morphological aspect of the ZTC sample is shown in the HRSEM high-resolution scanning electron microscopy images (secondary electron images—SEI), as shown in Figure 1a–d:

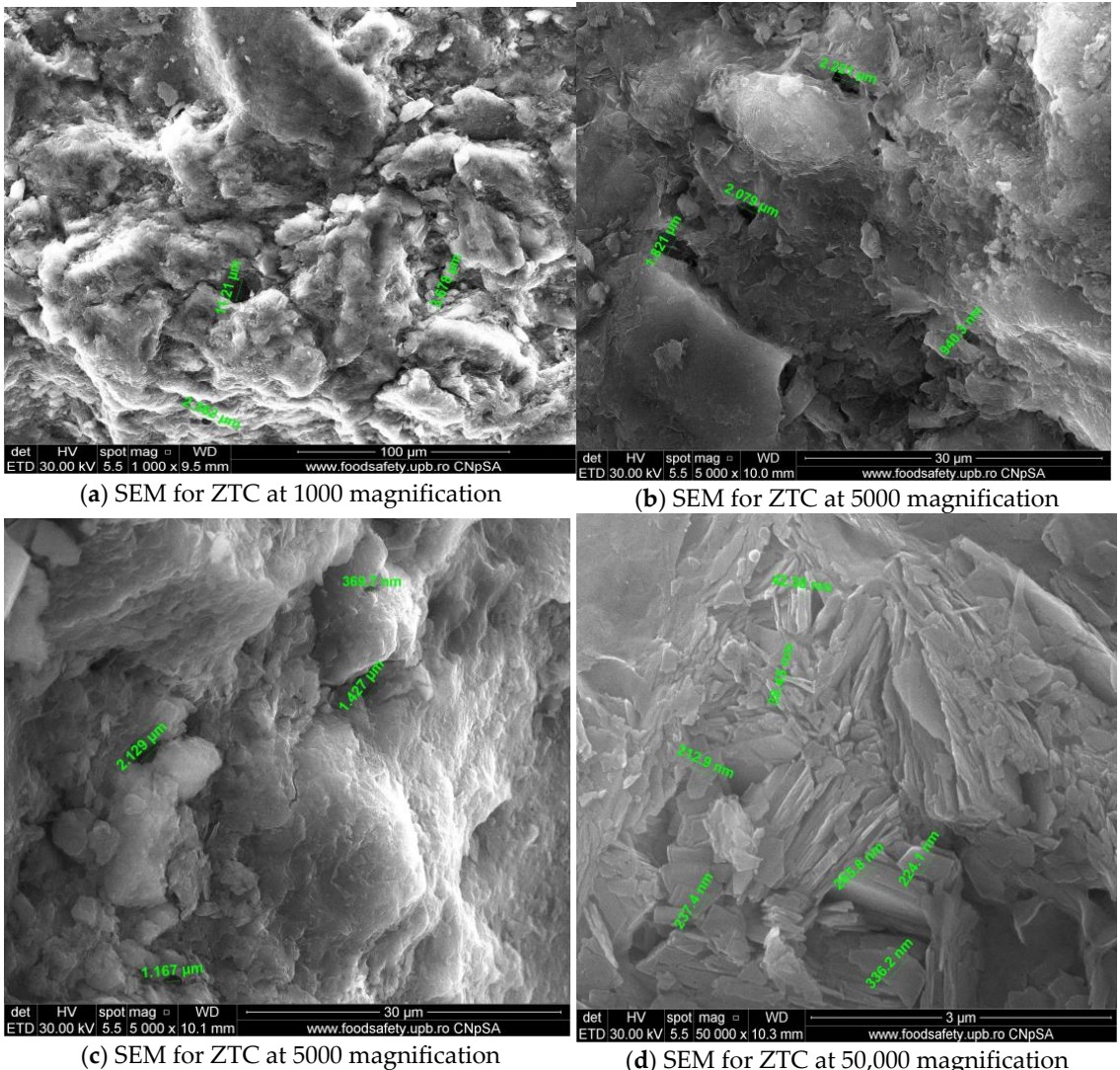

(**a**) SEM for ZTC at 1000 magnification

(**b**) SEM for ZTC at 5000 magnification

(**c**) SEM for ZTC at 5000 magnification

(**d**) SEM for ZTC at 50,000 magnification

**Figure 1.** (**a**–**d**) The SEM analysis for ZTC at different magnification.

From a microstructural point of view, regarding the pore size from SEM analysis, as shown in Figure 1a–d, a lamellar structure can be observed with platelet thicknesses of up to 100 nm. The pore size varies from 30 nm to 11 µm, which proves that ZTC can adsorb a very large range of compounds from water, including BPA. The EDAX spectrum highlighted the presence of the following main elements: O, Mg, Al, Si, K, Ca, and Fe. The high content of silicon, in an atomic ratio of Si/Al $\geq$ 4, can be seen in Figure 2 and Table 2.

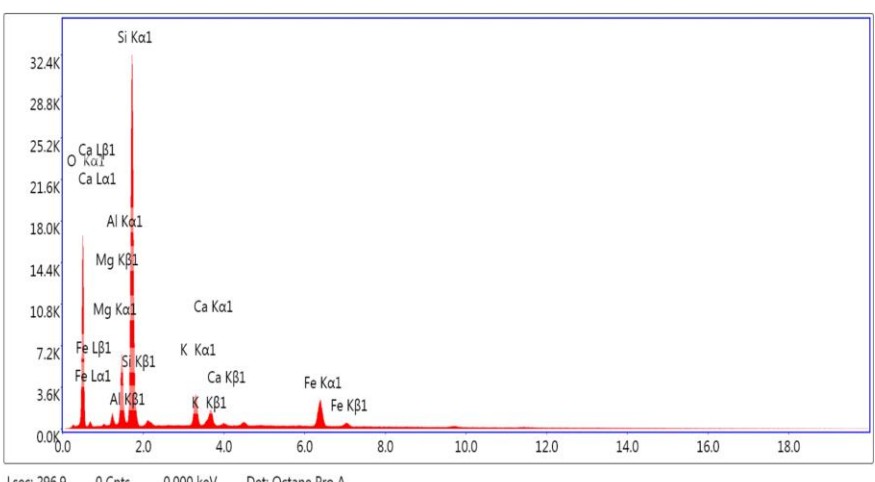

**Figure 2.** EDAX spectrum that is specific to ZTC.

**Table 2.** Elemental composition, weight, and atomic percentage of the component elements of ZTC.

| Element | Weight % | Atomic % |
| --- | --- | --- |
| O K | 50.40 | 65.81 |
| MgK | 1.37 | 1.18 |
| AlK | 6.77 | 5.24 |
| SiK | 30.85 | 22.95 |
| K K | 3.45 | 1.84 |
| CaK | 2.11 | 1.10 |
| FeK | 5.05 | 1.89 |

The distribution of the elements in the internal structure of the zeolites is highlighted in Figure 3. From the analysis of the EDAX spectrum, it can be observed that 46% of the image content was unallocated at 23,609 Pixels, 20% of the image content showing Si, O, Al, Fe, and K can be seen at a resolution of 10,033 Pixels, and 34% of the image containing Si and Al can be seen at a resolution of 17,558 Pixels. In Figure 3a, one can see the SEM image of the ZTC that was subjected to analysis. A more prominent area is identified, which can be associated with a crystallized area. The overall image of the distribution of all the component elements is presented in Figure 3b. One can observe a virtually uniform distribution of some elements that are present in the zeolite structure, such as Si, Al, Ca, and K (see Figure 3c–e,i), but we can also observe some agglomerations of elements, such as Fe and Mg in certain areas (Figure 3f,j). The latter elements are concentrated precisely in that area where the agglomeration appears.

### 3.1.2. XRD Analysis of ZTC

The diffractograms are presented in Figure 4.

In the case of sample P1, one can see the characteristic diffraction maxima of Ca Clinoptilolite zeolite $((Na_{1.32}K_{1.28}Ca_{1.72}Mg_{0.52})(Al_{6.77}Si_{29.23}O_{72})(H_2O)_{26.84})$, which is identified using PDF file 00-089-7535, accompanied by Phlogopite-1M $(KMg_3Si_3AlO_{10}(OH)_2)$, which is identified using PDF file 00-010-0495. It was also possible to highlight the presence of Albite $((Na_{0.84}Ca_{0.16})Al_{1.16}Si_{2.84}O_8)$, identified using PDF file 01-076-0927, as well as traces of crystalline $SiO_2$, identified using PDF file 04-020-9990. The sample marked P0 shows the same mineral phases identified above.

Following Rietveld processing, the proportion of phases could be highlighted, and the results are centralized in Table 3.

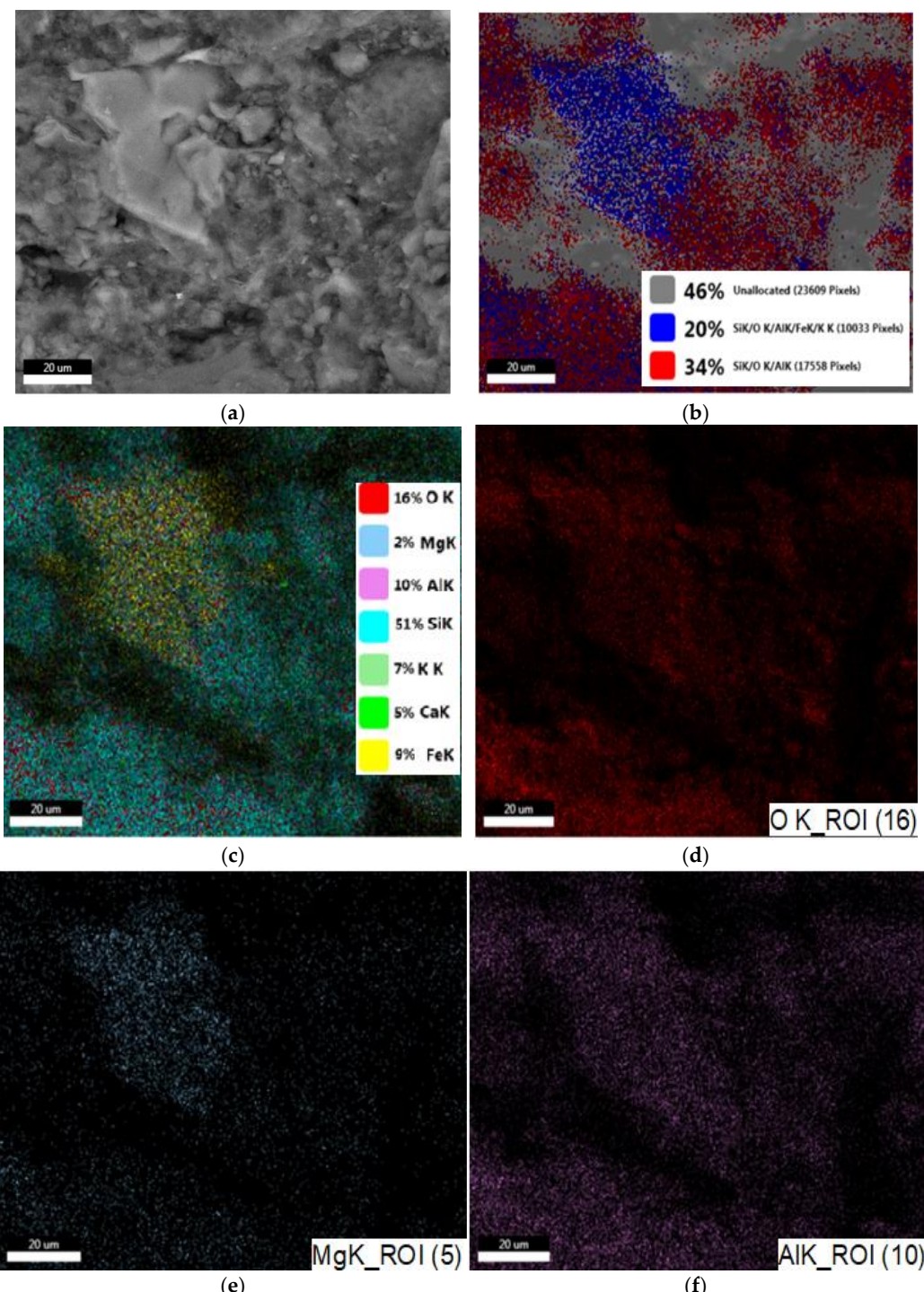

**Figure 3.** *Cont.*

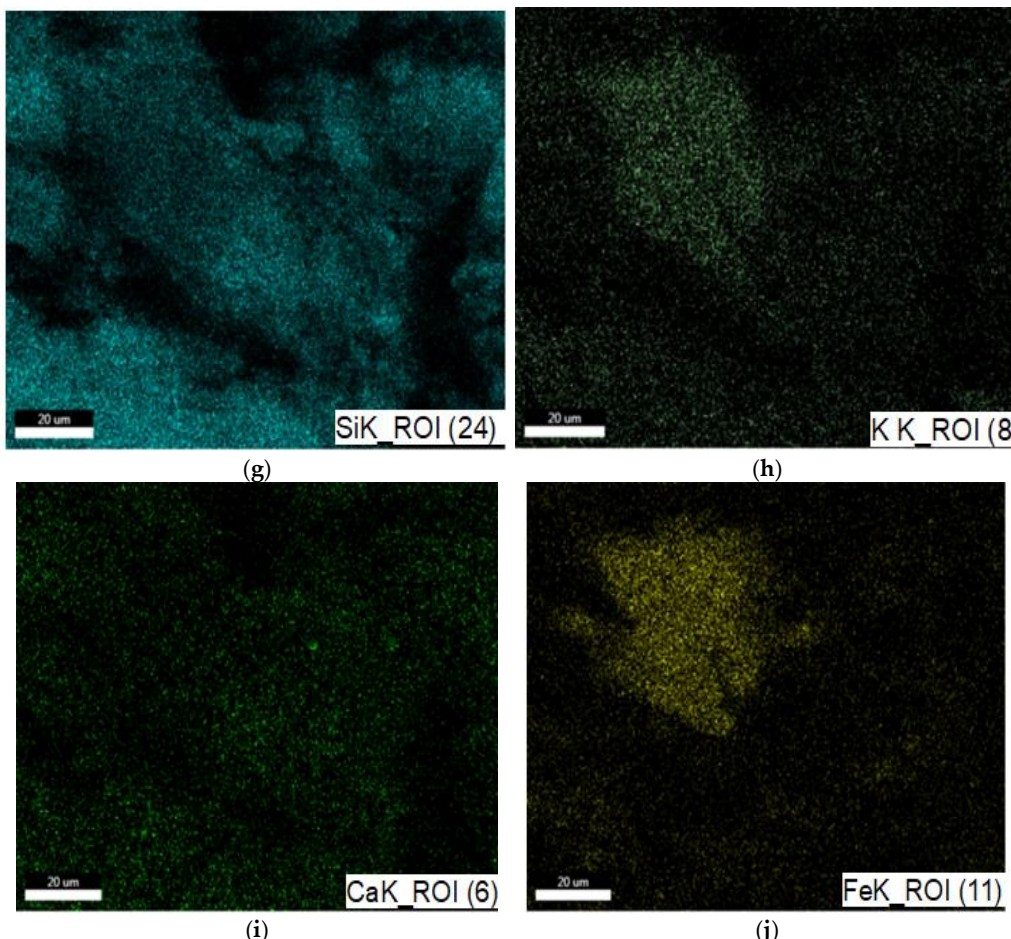

**Figure 3.** The EDAS distribution of elements in the internal structure of the ZTC. (**a**) General internal structure of the ZTC; (**b**) Unallocated and allocated percentage of the general internal structure of the ZTC; (**c**) General internal distribution of all the elements and its ratio for ZTC; (**d**) Internal distribution of oxygen in the internal structure of ZTC; (**e**) Internal distribution of magnesium in the internal structure of ZTC; (**f**) Internal distribution of aluminum in the internal structure of ZTC; (**g**) Internal distribution of silicon in the internal structure of ZTC; (**h**) Internal distribution of kalium in the internal structure of ZTC; (**i**) Internal distribution of calcium in the internal structure of ZTC; and (**j**) Internal distribution of iron in the internal structure of ZTC.

**Table 3.** Proportion of crystals in ZTC.

| No. | Compound | Chemical Formula | PDF File | Quantity (%) | |
|-----|----------|------------------|----------|------|------|
| | | | | P0 | P1 |
| 1 | Clinoptilolite Ca | $(Na_{1.32}K_{1.28}Ca_{1.72}Mg_{0.52})(Al_{6.77}Si_{29.23}O_{72})(H_2O)_{26.84}$ | 00-089-7535 | 80.3 | 80.8 |
| 2 | Phlogopite-1M | $(KMg_3Si_3AlO_{10}(OH)_2)$ | 00-010-0495 | 9.7 | 11.7 |
| 3 | Albite | $(Na_{0.84}Ca_{0.16})Al_{1.16}Si_{2.84}O_8$ | 01-076-0927 | 8.3 | 5.2 |
| 4 | Quartz | $SiO_2$ | 04-020-9990 | 1.7 | 2.3 |

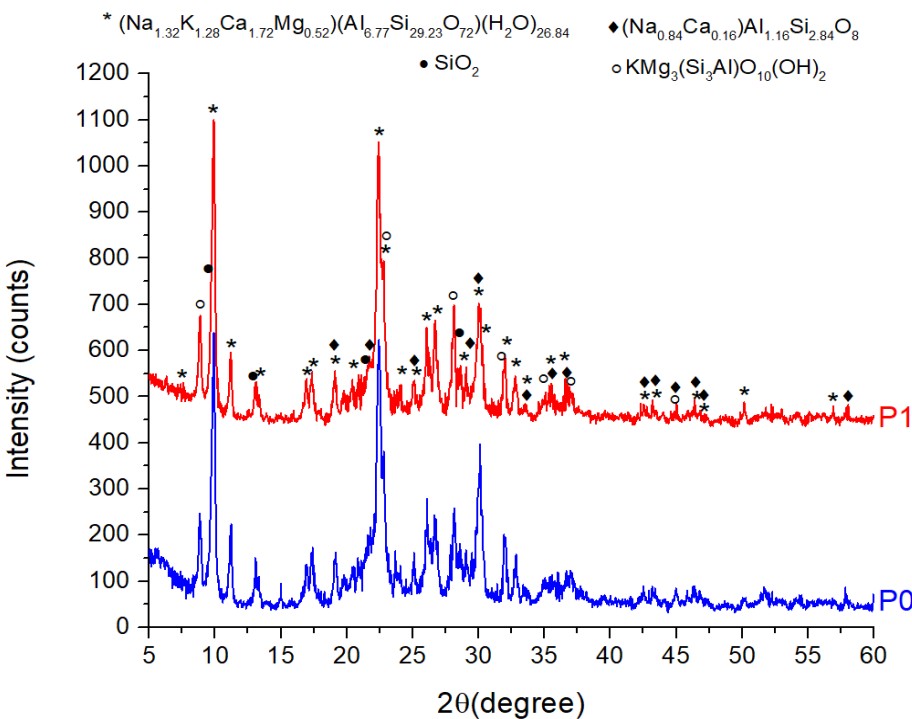

**Figure 4.** Characteristic ZTC diffractograms for two samples P0 and P1.

*3.2. Adsorption Equilibrium*

3.2.1. Adsorption Isotherm for Different pH Values

The isotherms for GAC and ZTC at pH = 8.12 are shown in Figure 5.

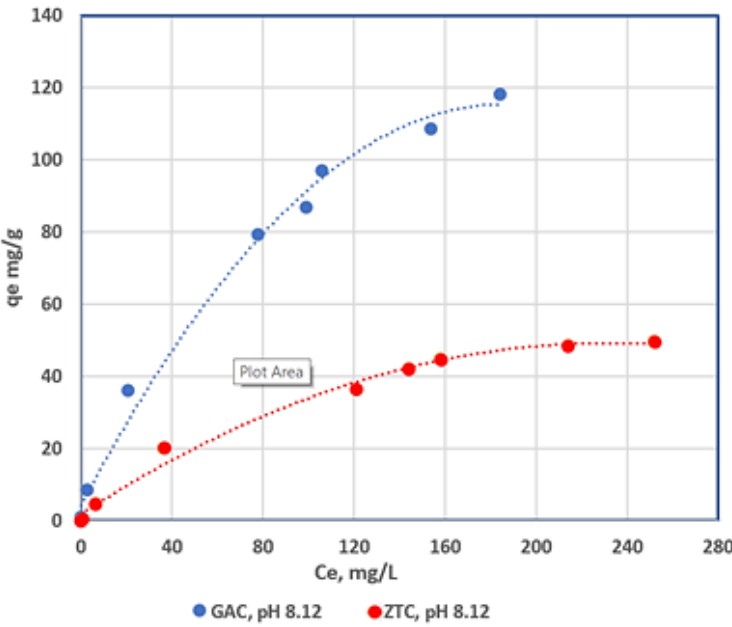

**Figure 5.** Adsorption isotherm for GAC and ZTC at 25 °C, I = 0 M, and pH = 8.12.

As expected, activated carbon has a much higher adsorption capacity for BPA than zeolite. However, the adsorption capacity is not the only criterion that must be taken into account when choosing an adsorbent. We must also take into account the consumption of raw materials, energy, water, the carbon footprint, the duration of the life cycle, and finally, cost.

The carbon footprint in the production of activated carbon is very large, given the fact that it is obtained from plant material or coal through heat treatment at 800–900 °C, for 2–10 h. Therefore, we have a high consumption of energy, and only 25% of the mass used results in the production of activated carbon; the rest of the mass produced are gases that reach the atmosphere, carbon dioxide being the main component. The duration of the life cycle for the stage wherein activated carbon is used can last between several months to several years; each reactivation stage comprises additional costs and energy consumption. Zeolite is a mineral adsorbent, which does not require special preparation techniques, it has a much longer lifetime (it can last for decades), and it has a much higher level of mechanical resistance. The desorption of organic compounds takes place at moderate temperatures without considerable mass loss. For drinking water purification, the market price for activated charcoal is ∼1200 EUR/ton [30], whereas the price of zeolite does not exceed 200 EURO/t.

Taking into account the abovementioned factors, it seems that the zeolite option is worth being taken into consideration.

The isotherms of the reaction were shown for three different pH values, for both adsorbents, are as follows: pH = 4.11, 6.98, and 8.12 (Figure 6a,b). Langmuir and Freundlich adsorption isotherms were performed to explain BPA adsorption. They explain the interactions between the adsorbent molecules and the adsorbate molecules, as well as the adsorption capacity. Adsorption can occur in monolayers or multilayers.

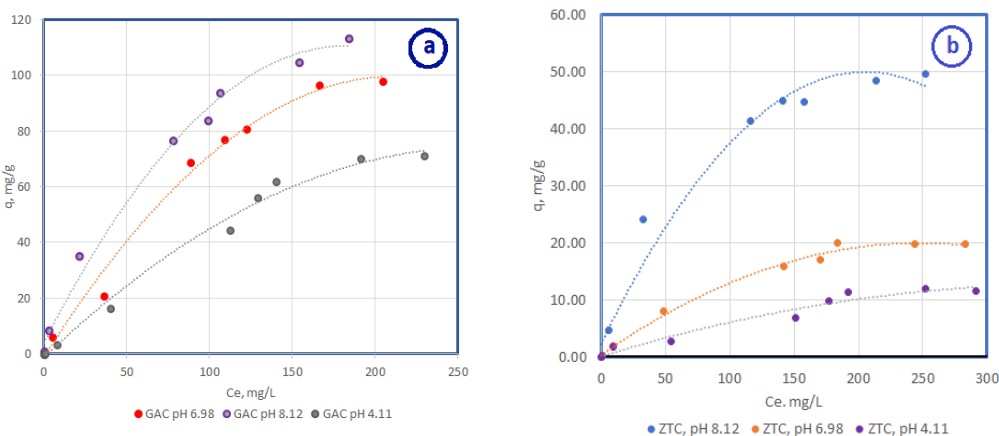

**Figure 6.** Adsorption isotherms at 25 °C, with an ionic strength of 0 M for (**a**) GAC and (**b**) ZTC at three pH values 4.11, 6.98, and 8.12.

The maximum adsorption capacity for both adsorbants increased as the pH increased for GAC, at values of 71 mg/g, 98 mg/g, and 113 mg/g, and for ZTC, at values of 50 mg/g, 20 mg/g, and 12 mg/g, at pH values of 4.11, 6.98, and 8.12, respectively.

The increase is less significant in the acidic pH range, almost imperceptible, for ZTC, after which, it increases suddenly in the basic range. It seems that the affinity between GAC and ZTC for the proton is much higher. The active centers which have previously been occupied with protons reject the BPA molecules, thus reducing the adsorption capacity in the acidic media. As a phenolic compound, when BPA is exposed to a high pH, it dissociates and forms hydrophobic anions.

For identifying the mechanism of the adsorption process, the experimental data were compared with the Langmuir and Freudlich mathematical models.

In Tables 4–6, the mathematical equations that are characteristic of Langmuir and Freudlinch models for BPA on GAC and ZTC are shown, and the values of the specific parameters are shown, were determined using mathematical models.

**Table 4.** The characteristic linearized equations, according to the Langmuir and Freudlinch models for BPA on GAC.

| pH | Linear Langmuir Equations | Linear Freudlinch Equations |
|---|---|---|
| 4.11 | $\frac{1}{q_e} = 2.0223\frac{1}{Ce} + 2.6403$ | $\log q_e = 0.5830 \log C_e + 1.0567$ |
| 6.98 | $\frac{1}{q_e} = 1.196\frac{1}{Ce} + 2.3293$ | $\log q_e = 0.5582 \log C_e + 0.9144$ |
| 8.12 | $\frac{1}{q_e} = 0.29\frac{1}{Ce} + 2.0039$ | $\log q_e = 0.5852 \log C_e + 0.4713$ |

**Table 5.** The characteristic linearized equations, according to the Langmuir and Freundlich models for BPA on ZTC.

| pH | Linear Langmuir Equations | Linear Freundlich Equations |
|---|---|---|
| 4.11 | $\frac{1}{q_e} = 4.181\frac{1}{Ce} + 19.253$ | $\log q_e = 0.7661 \log C_e + 1.3215$ |
| 6.98 | $\frac{1}{q_e} = 1.8256\frac{1}{Ce} + 10.045$ | $\log q_e = 0.9063 \log C_e + 1.0016$ |
| 8.12 | $\frac{1}{q_e} = 1.2897\frac{1}{Ce} + 3.4878$ | $\log q_e = 1.0309 \log C_e + 0.4886$ |

**Table 6.** The values of the equilibrium parameters, $q_m$, $b$ and $R^2$, which are characteristic of the Langmuir equation, and $K$, $n$, and $R^2$, respectively, and the characteristics of the Freundlich equation for BPA cations on GAC and ZTC.

| Adsorbent | Langmuir Isotherm | | | Freundlich Isotherm | | |
|---|---|---|---|---|---|---|
| | $q_m$, mg/g | $b$ | $R^2$ | $K$ | $n$ | $R^2$ |
| CA pH 4.11 | 84.5 | 1.3 | 0.9996 | 11.37 | 1.71 | 0.9064 |
| CA pH 6.98 | 95.78 | 1.94 | 0.9996 | 8.21 | 1.79 | 0.9588 |
| CA pH 8.12 | 111.34 | 6.91 | 0.9965 | 2.96 | 1.70 | 0.9392 |
| Zeolite pH 4.11 | 11.60 | 4.6 | 0.9961 | 20.96 | 1.3 | 0.9701 |
| Zeolite pH 6.98 | 22.21 | 5.5 | 0.9992 | 10.04 | 1.1 | 0.9711 |
| Zeolite pH 8.12 | 63.9 | 2.7 | 0.9995 | 10.73 | 0.97 | 0.9147 |

As is evident from Table 6, the experimental data for both GAC and ZTC, and the $R^2$ coefficients, have values above 0.99 for the Langmuir mathematical model in all cases studied. In the case of the Freundlich model, The $R^2$ coefficients were between 0.9064 and 0.9711. The experimental data show a linearity that is compatible with the Langmuir isotherm, indicating that the Langmuir mathematical model most faithfully describes the adsorption equilibrium of BPA. Similar results were obtained in a different study [31].

3.2.2. The Influence of Ionic Strength on the Adsorption Process

The presence of potassium chloride ions in higher or lower concentrations influences the adsorption process, as seen in Figure 7a,b.

Regarding GAC, we can observe that at a pH of 8.12, the adsorption capacity decreases by 4.2% at an ionic strength of 0.01 M KCl, and by 20% at an ionic strength of 0.1 M KCl. As the pH decreases, it can be observed that the competition for active centers increases; protons also intervene, which determines a more pronounced decrease in the maximum adsorption capacity. Ionic strength influences the adsorption capacity at 0.01 M, and it represents 91.5% and 96.7l%. At an ionic strength of 0.1 M it represents 68% and 73% of the initial capacity at pH values of 6.98 and 4.11, respectively.

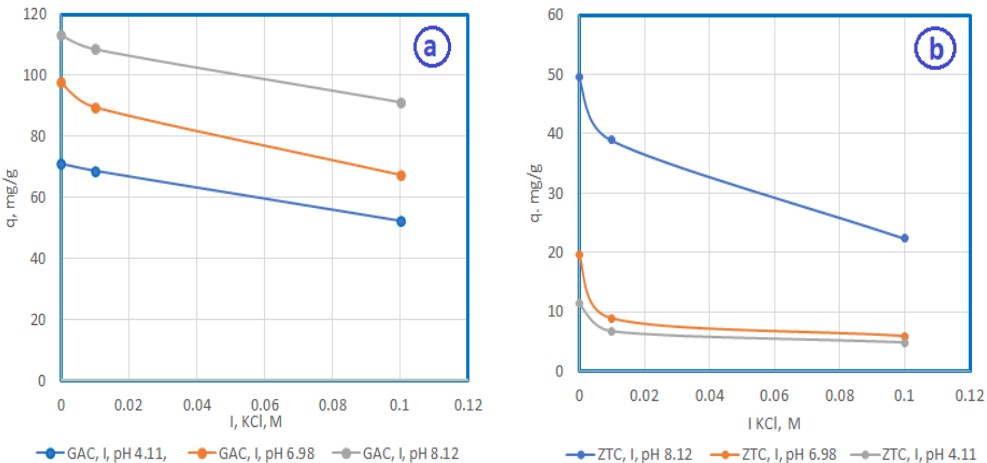

**Figure 7.** The variation in the maximum adsorbtion capacity of BPA for ionic strengths of KCl at 0, 0.01, and 0.1 M for (**a**) GAC and (**b**) ZTC at pH values of 4.11, 6.98, and 8.12.

Regarding ZTC, we can see that the present ionic strength affects the maximum adsorption capacity to a greater extent than activated carbon, and this is accentuated as the pH decreases. It can be observed that at an ionic strength of 0.01 M, the maximum adsorption capacity decreases to 79%, 30%, and 58% of the initial capacity, and at an ionic strength of 0.1 M, it reaches 42%, 45.2%. and 45%. At pH values of 4.11, 6.98, and 8.12, respectively. Unlike GAC, zeolite has an important ion exchange capacity, therefore, a large proportion of the active centers react with potassium cations, which then occupy the active centers. Activated carbon does not have an important adsorption capacity for potassium cations, and therefore, their effect is much smaller.

## 4. Discussion

BPA is an endocrine disruptor that has negative effects on humans and living organisms, therefore, it needs to be removed from domestic and industrial wastewater. Among the BPA removal technologies, as well as traditional and advanced methods, adsorption can satisfactorily remove this pollutant, because it is an effective method which is low cost, it is environmentally friendly, it uses a wide range of adsorbents with high reusability, and it is easy to operate. The adsorbents used in this study are as follows: activated carbon and Rupea zeolite. The adsorbents showed a good BPA removal capacity.

From specialized studies, it was highlighted that pure adsorbents demonstrated a lower adsorptivity compared with chemically or functionally modified adsorbents. In future research, the adsorption of BPA on adsorbents that were developed with modifications using different functional groups could be studied while monitoring the possible secondary effects of pollution, which is generated due to the added reactive substances. Another aspect that must be taken into account is the loss of adsorption sites via adsorbents after several cycles of the adsorption process, and their transformation into waste that generates secondary pollution.

## 5. Conclusions

Volcanic tuff zeolite from Rupea is a mineral that is found in extensive deposits in Romania, and the mining costs are very low. As mineralogical composition, it is made up of minerals of Clinoptilolite in a proportion of over 80%; it has a ratio of Si to Al that is greater than 4, which explains the high adsorption capacity for different ionic and molecular species. It has a uniform distribution of the main elements which comprise its composition, but there are also crystalline agglomerations. The pore size significantly varies, from a few tens of nm to tens of micrometers. This proves their dimensional compatibility for a very large range of components. Even if clinoptilolite is recognized for its high ion exchange capacity, studies have shown that it can also retain organic compounds such as BPA. This

retention mechanism is very well described using the Langmuir mathematical model. This mathematical model also describes the adsorption of BPA on GAC very well.

As per the studies that were carried out, it is evident that GAC has a BPA adsorption capacity of over 70 mg/g at an acidic pH, and it tends to have a capacity of 115 mg/g at a slightly basic pH. ZTC has a maximum adsorption capacity of around 12 mg/g at a pH of 4.11, and it tends to have a capacity of 50 mg/g at a pH of 8.12. The ionic strength determines the reduction in the adsorption capacity for both GAC and ZTC; it is more pronounced for ZTC, under low pH conditions, and at an ionic strength of 0.1 M, it can decrease significantly to a maximum capacity of 5 mg/g. At a basic pH at the same ionic strength, the capacity can reach over 22 mg/g. If the ionic strength is at a capacity of 0, the adsorption capacity of zeolite can reach 50 mg/g, and that of GAC tends to 115 mg/g. When choosing the adsorbent, one must take into account its technical performance, in addition to its life cycle costs and carbon footprint. Although zeolite does not have a very high adsorption capacity, it does have good mechanical resistance, it has a much longer cycle life, a reduced carbon footprint, and its cost is six to seven times lower than GAC.

**Author Contributions:** Conceptualization, D.S.S. and M.S.; methodology, D.S.S.; software, D.S.S.; validation, D.S.S., F.L.C. and M.S.; formal analysis, A.M.D.; investigation, A.M.D., R.T., A.I.N. and F.L.C.; resources, D.S.S.; writing—original draft preparation, D.S.S. and M.S.; writing—review and editing, D.S.S.; visualization, A.I.N. and R.T.; supervision, D.S.S.; project administration, A.M.D.; funding acquisition, D.S.S. and A.M.D. All authors have read and agreed to the published version of the manuscript.

**Funding:** This work has been funded by the European Social Fund from the Sectoral Operational Programme Human Capital 2014–2020, through the Financial Agreement with the title "Training of PhD students and postdoctoral researchers in order to acquire applied research skills—SMART", Contract no. 13530/16.06.2022—SMIS code: 153734.

**Institutional Review Board Statement:** Not applicable.

**Informed Consent Statement:** Not applicable.

**Data Availability Statement:** No new data were created.

**Acknowledgments:** We thank to the company, Zeolites Production, Rupea, Romania for supplying us the zeolite used in the studies.

**Conflicts of Interest:** The authors declare no conflict of interest.

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
