# Peer review of "Clinoptilolite—A Sustainable Material for the Removal of Bisphenol A from Water"

_sustainability, doi:10.3390/su151713253_

Round 1

Reviewer 1 Report

The manuscript entitled "Sustainability-2552870: Clinoptilolite - a sustainable material for removal of Bisphenol A from water", is based on the isothermal modelling of Bisphenol A removal from water. The suggested study is informative and can contribute the readers. I think, it may be accepted after a major revision.

-        In abstract, the information related to Bisphenol A should be moved to introduction for a better understanding.

-        The manuscript should be checked for the grammar mistakes by a native speaker. For instance,

Are the BFA and BPA are different or miswritten abbreviations?

In Table 4 and 5, “liniar” should be “linear”.

The subscript forms of the notations should be edited in adsorption results.

IN page 10 line349, “mathematique/cal ecuation” should be “mathematical equation”.

-        In SEM results, Figure 1 is not clear.

-        In Page 8 line 14, the figure caption was miswritten. Figure 1 and 2 should be edited as Figure 4.

-        For the correct interpretation of Langmuir isothermal modelling, the calculation of the separation factor (RL) is notable. The Rl values of the model should be added. For a better understanding in the interpretation the related articles could be added to the references.

Senberber, F. T., Yildirim, M., Mermer, N. K., & Derun, E., (2017). Adsorption of Cr(III) from Aqueous Solution using Borax Sludge. ACTA CHIMICA SLOVENICA , vol.64, no.3, 654-660.

H. S. Y. Akrawi et al 2021 IOP Conf. Ser.: Earth Environ. Sci. 761 012017

The manuscript should be checked for the grammar mistakes.

Reviewer 2 Report

As the author mentioned, with the development of society, the emergence of new pollutants in water treatment has brought new challenges, however, it has also attracted widespread attention from researchers. A large number of studies have developed different adsorbents and adopted various modification methods to improve their adsorption capacity for these pollutants. But the author did not have a clear understanding and grasp of the current research status, resulting in the novelty and significance of this study not being reflected.  In addition, there are still some problems in the manuscript, such as blurred figures, disordered tables, too many paragraphs, and the logic of writing the paper is confused. It is suggested that the author carefully revise and submit the manuscript.

No.

Reviewer 3 Report

In this manuscript, the influence of different PH values on the adsorption performance of zeolite is analyzed, and the adsorption mechanism is discussed. In general, the authors have made efforts to find effective adsorbents that can replace activated carbon. However, the evidence for the properties and advantages described in the manuscript is insufficient. The authors should supplement this appropriately. In addition, there are numerous errors in the article and I suggest that the authors carefully check and correct it. Specific questions are as follows:

1.       How was the pore size of ZTC obtained?

2.       The description of Figure 3 in the manuscript does not correspond to that on the pictures and should be carefully examined.

3.       What do samples P0 and P1 represent, respectively? Samples P0 and P1 were not defined in the manuscript.

4.       There are many careless errors in the manuscript. For example, the Mn element mentioned in line 286 is not present in the sample. The sequence number of the diffraction pattern is wrong in line 291, etc. Authors should carefully check the entire manuscript.

5.       The authors emphasize that zeolite has a much longer cotton life, how is it proved? In addition, does the zeolite have good cycle stability for the adsorption of bisphenol A?

Minor editing of English language required

Round 2

Reviewer 1 Report

The revised version of the manuscript is much more suitable for acceptance.

Moderate editing of English language required